# Metabolic Footprint, towards Understanding Type 2 Diabetes beyond Glycemia

**DOI:** 10.3390/jcm9082588

**Published:** 2020-08-10

**Authors:** Ana F. Pina, Rita S. Patarrão, Rogério T. Ribeiro, Carlos Penha-Gonçalves, João F. Raposo, Luís Gardete-Correia, Rui Duarte, José M. Boavida, José L. Medina, Roberto Henriques, Maria P. Macedo

**Affiliations:** 1CEDOC-Centro de Estudos de Doenças Crónicas, NOVA Medical School, Faculdade de Ciências Médicas, Universidade Nova de Lisboa, 1150-082 Lisboa, Portugal; ana.pina@nms.unl.pt (A.F.P.); rita.patarrao@nms.unl.pt (R.S.P.); filipe.raposo@apdp.pt (J.F.R.); 2ProRegem-ProRegeM PhD Programme, NOVA Medical School/Faculdade de Ciências Médicas, Universidade NOVA de Lisboa, 1150-082 Lisbon, Portugal; 3IGC-Instituto de Gulbenkian e Ciência, 1150-082 Lisbon, Portugal; cpenha@igc.gulbenkian.pt; 4Portuguese Diabetes Association-Education and Research Center (APDP-ERC), 1150-082 Lisbon, Portugal; rogerio.ribeiro@apdp.pt (R.T.R.); luisgardete@netcabo.pt (L.G.-C.); ruisduarte3@gmail.com (R.D.); jose.boavida@apdp.pt (J.M.B.); 5SPD-Sociedade Portuguesa de Diabetologia, 1150-082 Lisbon, Portugal; jlmedina40@gmail.com; 6NovaIMS-Nova Information Management School, 1150-082 Lisbon, Portugal; roberto@novaims.unl.pt

**Keywords:** diabetes, heterogeneity, clustering, dysmetabolism

## Abstract

Type 2 diabetes (T2D) heterogeneity is a major determinant of complications risk and treatment response. Using cluster analysis, we aimed to stratify glycemia within metabolic multidimensionality and extract pathophysiological insights out of metabolic profiling. We performed a cluster analysis to stratify 974 subjects (PREVADIAB2 cohort) with normoglycemia, prediabetes, or non-treated diabetes. The algorithm was informed by age, anthropometry, and metabolic milieu (glucose, insulin, C-peptide, and free fatty acid (FFA) levels during the oral glucose tolerance test OGTT). For cluster profiling, we additionally used indexes of metabolism mechanisms (e.g., tissue-specific insulin resistance, insulin clearance, and insulin secretion), non-alcoholic fatty liver disease (NAFLD), and glomerular filtration rate (GFR). We found prominent heterogeneity within two optimal clusters, mainly representing normometabolism (Cluster-I) or insulin resistance and NAFLD (Cluster-II), at higher granularity. This was illustrated by sub-clusters showing similar NAFLD prevalence but differentiated by glycemia, FFA, and GFR (Cluster-II). Sub-clusters with similar glycemia and FFA showed dissimilar insulin clearance and secretion (Cluster-I). This work reveals that T2D heterogeneity can be captured by a thorough metabolic milieu and mechanisms profiling—*metabolic footprint*. It is expected that deeper phenotyping and increased pathophysiology knowledge will allow to identify subject’s multidimensional profile, predict their progression, and treat them towards precision medicine.

## 1. Introduction

It is widely accepted that type 2 diabetes is a multifactorial and heterogeneous condition progressing through multiple pathophysiological mechanisms [1]. Despite recent therapeutic advances and enormous expenditure in treatment [2] type 2 diabetes carries a heavy socioeconomic burden [3] and is a leading cause of renal failure, blindness, cardiovascular events, and major amputations of the lower limbs [4]. Current type 2 diabetes diagnosis criteria and therapeutic targets are mainly focused on glycemia and Hemoglobin A1c (HbA1c) measurements, but improvements in health outcomes demand tools that rationally stratify the different states of dysglycemia. In response to these needs, cluster analysis, a powerful data mining tool, allows exploration and rationalization of complex data, through stratification into groups, maximizing similarity within the group and minimizing it between groups [5].

Recently, Alhqvist et al. [6], used cluster analysis to stratify patients diagnosed with diabetes. For this they chose six parameters to inform the cluster, namely insulin resistance, insulin secretion, age at diagnosis of diabetes, body mass index (BMI), HbA1c and autoantibodies. The authors identified five clusters, four of them mainly corresponding to type 2 diabetes. Importantly, this clustering profile was replicated in other cohorts [7,8] and revealed a higher prevalence of nephropathy in a cluster characterized by severe insulin resistance (SIR), and retinopathy in a cluster with severe insulin deficiency (SID). Nevertheless, it has been suggested that these clusters do not perform better than simple parameters such as age at diagnosis, gender, and kidney function in predicting complications and therapeutic responses [7]. As early as in 1965, the World Health Organization (WHO) recognized two main groups in type 2 diabetes: insulin resistant and insulin deficient subjects [9]. Since then different organs, such as liver, gut, adipose tissue, and the central and peripheral nervous system were implicated in the pathophysiology of dysglycemia, through distinct mechanisms [10].

In 2017, Mark McCarthy proposed a conceptual framework for diabetes diagnosis, the “palette” model [11]. In this model, dysglycemic mechanisms and traits sum up in a phenotype and map the subject in the path from normoglycemia to diabetes. Nonetheless, there are people diagnosed with type 2 diabetes, who do not develop complications, whereas people with prediabetes may already have complications [12]. In fact, besides blood glucose levels, people are exposed to other important metabolic components (e.g., lipids, insulin), herein, referred to as metabolic milieu.

In line with the “palette” concept, we propose that pathophysiological mechanisms of metabolic diseases quantitatively determine clinical parameters, related to glucose, insulin, and free fatty acids metabolism, explaining distinct dysmetabolic profiles. Using a cluster analysis, we aimed at stratifying a population regarding the metabolic milieu, including subjects with normoglycemia and dysglycemia, but without any prior assumption on classical parameters thresholds or cut-offs. This stratification was considered to reveal chief pathophysiological mechanisms involved in dysmetabolism, therefore allowing differential therapeutic management of glycemic states towards personalized interventions.

## 2. Experimental Section

### 2.1. Ethics Statement

All subjects were volunteers and provided written informed consent for participation in this study. Ethical permits to conduct this study were obtained from the Ethics Committee of Portuguese Diabetes Association (APDP-Diabetes Portugal). The study protocol adhered to the Declaration of Helsinki and was approved by the Data Protection Agency (permit nr.3228/2013).

### 2.2. Subjects

The study population comprises participants of a diabetes prevalence study performed in Portugal (PREVADIAB2). Between 2008 and 2009, the PREVADIAB1 study recruited 5167 subjects attending the national health care system across the country that were screened for diabetes status [13]. Normoglycemia, prediabetes, and diabetes were defined using the International Diabetes Federation (IDF)/World Health Organization (WHO) glycemia criteria [14,15]. In 2014, we selected within this cohort 1088 subjects that did not have diabetes in PREVADIAB1 and attended 55 health units evenly representing the geographical distribution of the Portuguese population (PREVADIAB2; Figure 1). For all participants, a letter was sent containing information for study participation.

### 2.3. Clinical Measurements

Clinical history of the participants in PREVADIAB2 was assessed, including anti-diabetic medication. Participants were weighed on an electronic scale, to the nearest 0.01 kg while wearing minimal clothes and without shoes. Waist circumference was measured with a flexible ruler, at the midline between the top of the iliac crest and the inferior limit of the ribs [16]. Height was measured to the nearest 0.1 cm with a stadiometer according to the standardized WHO procedures. Blood collection was performed, in a seated position from the antecubital vein at rest, after an overnight fast, at 0, 30 and 120 min of the oral glucose tolerance test (OGTT). The blood was collected into dry tubes and into tubes containing ethylenediamine-tetraacetic acid (EDTA) as an anticoagulant. Biological samples were centrifuged at 3500 rpm at 4 °C for 10 min and plasma samples were frozen at −80 °C for posterior analysis. Serum samples were used to analyze the lipid profile of the participants, including total cholesterol, low density lipoprotein (LDL-c), high density lipoprotein (HDL-c) cholesterol, free fatty acids (FFA), and triglycerides (TG) and to quantify creatinine, gamaglutamiltranspeptidase (GGT), alanine aminotransferase (ALT), and aspartate aminotransferase (AST) using colored enzymatic tests in an automated analyzer (Auto Analyzer Olympus AU640, Beckman Coulter, Brea, CA, USA). Glycated hemoglobin was determined by high performance liquid chromatography (HPLC) with boronate affinity (Menarini Premier Hb 9210, Paço de Arcos, Lisboa, Portugal). Plasma glucose levels were measured with a glucose analyzer (glucose oxidase method, Olympus AU640, Beckman Coulter, Brea, CA, USA). Plasma insulin and C-peptide levels were determined by commercial chemiluminescence assays, Liaison (DiaSorin, Salugia, Pidemont, Italy).

### 2.4. Metabolic And Functional Indexes

To assess glucose metabolism, besides the glycemic values of three time points of the OGTT (0 min, 30 min and 120 min), we profiled the clusters with the area under the curve (AUC), calculated with the same glycemia values (AUC_(0–30–120)_Glucose) using the trapezoidal rule, and with the HbA1c. Similarly, we used the three time points values of the OGTT (0 min, 30 min, and 120 min) of FFA, insulin, and C-peptide, and calculated the corresponding AUC’s (AUC_(0-30-120)_FFA, AUC_(0–30–120)_Insulin, AUC_(0–30–120)_C-peptide, respectively). To estimate the beta-cell function we calculated the homeostatic model index HOMA-B2 (HOMA calculator (University of Oxford, Oxford, UK) [17]) for the fast period using the C-peptide values, and for the absorptive period the indexes were derived from the OGTT values, namely AUC_(0-30)_C-peptide/AUC_(0–30)_Glucose and AUC_(0–30)_Insulin/AUC_(0–30)_Glucose, by (C-peptide0min + C-peptide30min)/(Glucose0min + Glucose30min) and (Insulin0min + Insulin30min)/(Glucose0min + Glucose30min) [18], respectively, insulinogenic index (IGI) calculated as (Insulin30min – Insulin0min)/(Glucose30min–Glucose0min), and similarly Δ_(0–30)_C-peptide/Δ_(0–30)_Glucose as (C-peptide30min–C-peptide0min)/(Glucose30min–Glucose0min). Disposition Index (DI) was calculated as IGI/Insulin0min [19]. We used the HOMA-IR2, also using the HOMA calculator, to evaluate the insulin resistance at fasting state, and for the absorptive period we calculated Matsuda Index, with insulin in the three time points of the OGTT (0 min, 30 min, and 120 min) [19]. Additionally, we profiled the clusters with organ specific insulin resistance indexes, namely for the liver (Liver-IR and Hepatic-IR) and adipose tissue (Adipo-IR and ISI-FFA). Liver-IR [20] was calculated as − 0.091 + (log AUC_(0-120)_Insulin × 0.400) + (logfatmass% × 0.346) – (logHDL-cx0.408) + (logBMIx0.435), and Hepatic-IR [21] as AUC_(0–30)_Insulin × AUC_(0–30)_Glucose. Adipo-IR [22] was derived as FFA0min × Insulin0min, and ISI-FFA [23] as 2/((AUC_(0–30–120)_Insulin × (AUC_(0–30–120)_FFA) + 1). Insulin clearance was estimated through insulin clearance at fast (Fasting IC), calculated as C-peptide0min/Insulin0min, the suppression of insulin clearance from 0 min to 30 min of the OGTT (Δ_(0–30)_IC), derived by the slope of insulin clearance at fast and insulin clearance at 30 min, and additionally, by the insulin clearance at 120 min (IC 120 min). Insulin clearance at 30 min and 120 min were calculated by AUC_(0–30)_C-peptide/AUC_(0–30)_Insulin and AUC_(30-120)_C-peptide/AUC_(30-120)_Insulin, respectively. We used surrogate indexes to evaluate fatty liver, aiming at accurately detecting this condition [24]. NAFLD-FLS [25] was calculated as −2.889 + 1.179 × IDF metabolic syndrome(yes = 1/no = 0) + 0.454 × type2diabetes (yes = 2/no = 0) + 0.145 × Insulin0min (mIU/L), and FLI as (e^0.953xln(TG) + 0.139 × BMI + 0.053 × Waist circumference −15.745^)/(1+e^0.953 × ln(TG) + 0.139 × BMI + 0.053 × Waist circumference − 15.745^) × 100) [26]. Finally, we estimated the glomerular filtration rate (eGFR) using the Chronic Kidney Disease Epidemiology Collaboration (CKD-EPI) formula [27]. BMI was calculated as weight (kg)/height^2^(m). Formulas are summarized in Appendix A.

### 2.5. Statistical Analysis

Comparison between genders for parameters informing the cluster algorithm and profiling the clusters was performed with Mann–Whitney test for continuous variables and with chi-square test for categorical variables.

### 2.6. Cluster Analysis

To inform the clustering algorithm and stratify the population we chose age, BMI, waist circumference, glycemia, insulinemia, C-peptide, and FFA measurements at three time points (0 min, 30 min and 120 min of the OGTT). We reasoned that these measurements, available in the clinic, would reflect clinically relevant variations in the metabolic milieu and are sensitive descriptors of dysglycemia heterogeneity.

The methodology workflow used for clustering and cluster profiling is outlined in Figure 1. In the PREVADIAB2 cohort 58 subjects under anti-diabetes medication were excluded from analysis. Likewise, further 59 subjects with missing values on variables informing the clustering algorithm were also excluded. BMI and waist circumference were scaled by min–max standardization by gender. We did not exclude outliers. Odd values of measured parameters were confirmed by a repeated measurement and outliers may be part of the dysglycemia heterogeneity we aim to capture. Furthermore, cluster analysis is an appropriate tool to identify outliers on a multidimensional level.

We used a hierarchical self-organizing map followed by a hierarchical clustering algorithm to stratify the population [28]. This methodology has been used to solve several complex clustering problems, namely in clustering analysis of type 1 diabetes complications [29]. The first algorithm allows data dimensionality reduction. We reduced data dimensionality to 27 units, using an implementation of multilayer self-organizing mapping in R (kohonen package [30]); considering that there are three accepted classes for fasting glycemia and for glucose tolerance, and that we used three time points values of the OGTT (0 min, 30 min and 120 min), 27 are the possible combinations. We used five toroidal-grids, each of which with 27 hexagonal units. The first four included three time point values of the OGTT of glycemia, C-peptide, insulinemia, and FFA, respectively, while the fifth included age, BMI, and waist circumference. After tuning, the remaining algorithm hyperparameters were set as follows: distance metric used was Euclidean distance; the algorithm was randomly initialized, and trained sequentially (500 iterations); default learning rate was used, decreasing linearly from 0.05 to 0.01; Gaussian neighborhood function, starting with default 2/3 of units, and decreasing linearly. The algorithm was run 1000 times and the model with the lowest quantization error was selected.

The resulting 27 units were then clustered and explored with the hierarchical clustering algorithm implementation in R, using the Ward method (hclust function, stats package [31]), which allows to explore the aggregation of SOM units in a hierarchical manner using different granularity (cut levels). Optimal number of clusters for the 27 SOM units was determined with the silhouette width (factoextra package [32]). Cluster stability was assessed by running the described analysis on 100 bootstrapped resamples and calculating the mean Jaccard coefficient [33].

We profiled the clusters for the overall population as well as separately for both genders, by assigning the median of the variable of each cluster to their distribution in the overall population (percentiles 10th, 25th, 75th, and 90th). In profiling the clusters, we used several surrogate indexes of important pathophysiological mechanisms and traits, in addition to the parameters informing the clustering.

## 3. Results

We analyzed 974 subjects of the PREVADIAB2 Portuguese cohort with median age of 62 years and 60% women. According to IDF/WHO guidelines for dysglycemia, 72% had normoglycemia, 22% prediabetes, and 6% diabetes, with no gender difference (*p* = 0.08). Table 1 summarizes descriptive statistics of the variables informing cluster analysis and cluster profiling. BMI and waist circumference scaled by gender, age, and the three time points (0 min, 30 min and 120 min of OGTT) of glycemia, insulinemia, C-peptide, and FFA levels were chosen to inform the clustering procedure. We reasoned that these measurements can be obtained in clinical practice, reflect clinically relevant variations in the metabolic milieu and together are sensitive descriptors of dysglycemia heterogeneity. In addition, cluster profiling incorporated surrogate indices of relevant dysmetabolism mechanisms. We assigned the clusters median of the profiling variables to the corresponding percentile of their distribution in the overall cohort (10th, 25th, 75th, and 90th percentile). Overall, the 25th–75th interval encompassed the accepted normal values of the profiling variables.

Firstly, we reduced data dimensionality to 27 units, using a SOM algorithm (Appendix A). To resolve the phenotypic variation of the 27 SOM units we used a hierarchical clustering algorithm, capturing subject’s similarity at different hierarchical aggregation levels. After the training procedure we found that the optimal number of clusters was two (silhouette index) with cluster stability (mean Jaccard coefficient 0.8).

The first cutting level of hierarchical clustering of the 27 SOM units identified two clusters (Figure 2; K = 2), where Cluster-I (C-I) included the majority of subjects under analysis (77%) with high representation of normoglycemia. In C-I, the median of all profiling variables fell within the range of the 25th to 75th percentiles of the global population. In contrast, Cluster-II (C-II) showed higher BMI, waist circumference, glycemia, insulinemia, and C-peptide levels, and therefore is associated with insulin resistance and fatty liver (median in 75th–90th percentiles). An analysis by gender showed similar profiling patterns, though expected differences in some variables such as BMI and waist circumference were detected (Appendix A). We noticed that C-I had a lower proportion of subjects with glycemia but, interestingly their absolute count was higher than C-II (145 subjects in C-I vs. 122 in C-II). This indicates that both clusters contain a significant number of individuals with glycemia suggesting an underlying phenotypic heterogeneity which is not captured by the optimal number of clusters. Our analysis proceeded towards higher granularity to an intermediate hierarchical cut level (K = 11) and full cluster discrimination (K = 27).

The initial breakdown of C-I in three clusters (C–A, C–B, and C–C) reveals very different profiles (Figure 2; K = 11). While C–C represents ~60% of C-I with large representation of normoglycemia, subjects with dysglycemia were overrepresented in C–B and showed a distinct profile with impaired insulin secretion and increased insulin clearance. Further C–B discrimination (K = 27, C-1 to C-27) distinguished Clusters 6 (C-6) and 7 (C-7) that retained the combined insulin secretion and insulin clearance unbalances but are differentiated by the increased free fatty acids levels in C-7 but not in C-6 (Figure 3; Appendix A for males and females, respectively). These two clusters represent 9% of the all cohort and uncovered a distinct pathophysiological trajectory entailing impaired insulin secretion and increased insulin clearance that can be associated or not with lipid dysmetabolism. Cluster 8 (C-8) did not show unbalances of evaluated dysmetabolism mechanisms, nonetheless subjects were exposed to increased median levels of both glycemia and FFA. This profile may represent individuals that develop dysglycemia and dyslypidemia by mechanisms that are uncommonly assessed. In C–A (Figure 2; K = 11), the median levels of the evaluated mechanisms fell within the 25th and 75th percentiles of the overall population, except for insulin secretion. Nevertheless, C–A harbored a cluster (Cluster 4) representing 3% of the cohort with normal insulin secretion but with hyperinsulinemia, which is associated with insulin clearance suppression and fatty liver. Noteworthy, in this cluster, median glycemic levels were increased at 30 minutes (Figure 3; Appendix A for males and females, respectively).

In C-II we focused in three sub-clusters of hierarchical discrimination for K = 27 (Clusters 18, 22, and 24) that covered distinct glycemia/FFA profiles and considering their representativeness in the population (Figure 3; Appendix A for males and females, respectively). In Cluster 18 all subjects had normoglycemia and the median glycemia and FFA values fell within the middle range of the overall population. However, they were exposed to hyperinsulinemia, plausibly explained by decreased insulin clearance as well as increased beta-cell function, in agreement with increased DI. This profile may represent a dysmetabolic compensation of the observed insulin resistance, namely in liver, that associates with NAFLD.

Clusters 22 and 24 are similarly associated with NAFLD and decreased eGFR (Figure 3; Appendix A for males and females respectively). However, both clusters were exposed to hyperglycemia and hyperinsulinemia, nevertheless only C-24 was exposed to higher FFA levels while showing lower adipose tissue insulin resistance than C-22. Strikingly, higher adipose insulin resistance was uncoupled from increased FFA levels in C-22.

## 4. Discussion

In this work we analyzed a population-based cohort, comprising subjects with normoglycemia non-treated pre-diabetes, and previously undiagnosed diabetes, and we identified a substructure of two optimal clusters (C-I and C-II). C-II was characterized by high insulin resistance, hyperinsulinemia, and fatty liver, recapitulating an expected profile of type 2 diabetes (insulin resistance) that, unexpectedly, included already a significant proportion of normoglycemic subjects. Within C-I, we identified the other well-known subgroups of insulin deficiency, that likewise, included individuals with normo- and dysglycemia. Overall, this confirms that insulin resistance and insulin deficiency profiles occur across all currently accepted classes of dysglycemia and also in normoglycemic subjects.

Our analysis further revealed that the two distinct clusters are made up of heterogenous groups regarding both, mechanisms and the milieu, that importantly have been independently associated with the development of diabetes’ complications [34,35]. In fact, the optimal number of clusters is defined based on subjects’ distances in a hyperspace, that are not necessarily clinically significant. However, cluster analysis is also a useful exploratory tool and, therefore, we proceed the analysis to higher levels of granularity.

We found that subjects with similar glycemia disturbances fall in clusters that differ not only in respect to insulin secretion and overall insulin resistance, but also regarding tissue-specific insulin resistance and insulin clearance, as illustrated in Clusters 6 vs. 22 and 8 vs. 24 (Figure 4). Conversely, within many clusters we found subjects that, despite being currently classified as having different glycemic status including normoglycemia, show a similar pattern of pathophysiological mechanisms. Whether such subjects with normoglycemia are protected by some non-evaluated mechanisms or traits, or on contrary, they are already progressing towards diabetes complications, cannot be discerned at this point. Nevertheless, these results indicate that the standard clinical evaluation of fasting and post-prandial glycemia does not reflect the overall exposure to glucose or express the metabolic milieu heterogeneity and that it is not informative about mechanisms governing the metabolic status. This may partly explain why some people classified with prediabetes might develop complications at early stages and others don’t [12,36]. Likewise, we found clusters that, despite having the same fatty liver prevalence, can have or not dysglycemia (Clusters 18, 22, and 24). This might indicate diverse fatty liver etiology [37]. Curiously, the clusters with dysglycemia also had decreased glomerular filtration rate, supporting a role of the kidney in disease progression.

A previous cluster analysis of subjects with diabetes revealed five clusters, four of which were related to type 2 diabetes [6]. Beyond severe insulin resistance and insulin deficiency, Alqhvist et al. [6] identified one cluster related with age at onset of diabetes and another with BMI. We found a cluster (C-8), in which dysglycemia was not associated with insulin resistance or insulin deficiency, but also could not be explained by age or BMI. Therefore, this cluster may represent a dysglycemic trajectory that operates by mechanism yet to be uncovered.

In our study we focused on clusters with higher prevalence. However, we found that less prevalent clusters comprised mainly subjects with dysglycemia, despite the lower proportion of prediabetes and diabetes in our sample. The occurrence of these metabolic profiles, if validated in larger studies, may deserve attention in the context of the high prevalence of dysglycemia in the world. A limitation of our study is that we could only use cross-sectional data, which does not allow addressing the existence of dynamic paths across the clusters throughout time that could suggest effective preventive interventions. This should be assessed in future longitudinal studies. Furthermore, the high number of included parameters can be also considered a limitation. Analysis with fewer parameters might be more perceptible and thus more appealing. Nonetheless, our work suggests that such approach is not sufficient to reveal the metabolic complexity.

McCarthy proposed that different phenotypes of type 2 diabetes result from the combination of several mechanisms and traits [11]. Pathophysiological mechanisms that drive dysglycemia are affected in a continuous manner and can be differently combined. Of note, this model places the subject in a path from normalcy to disease. Our work, to the best of our knowledge, is the first to stratify a population including subjects with type 2 diabetes, prediabetes, and normoglycemia, not relying in the current diagnostic criteria. We found that metabolic heterogeneity can be captured by taking in account not only glycemia levels but also other components of the milieu, disclosing underlying pathophysiological mechanisms.

We could not address whether the founded clusters are associated with specific type 2 diabetes complications. We speculate that it might be the case, given that they are highly stratified in both milieu and mechanisms and we found clusters with higher fatty liver and others with lower estimated glomerular filtration.

Radar plots in Figure 4 illustrate the metabolic footprint of the clusters discussed above, where we can easily assess each milieu component (yellow), mechanisms affection (blue), as well as presence of pathological conditions (red). This representation supports the notion that glycemic imbalances is only one axis of a broader multidimensional profile. The most recent type 2 diabetes treatment guidelines involve a multifactorial treatment targeted to glycemic and other risk factors control, aiming to prevent chronic complications. The pharmacological armamentarium in diabetes includes new classes of drugs with different mechanisms of action in various organs. Still, many of the choices are based on trial and error methods or on the assessment of cardiovascular complications or kidney disease. A deeper understanding of the underlying mechanisms, taking in account the milieu beyond glycemia, and therefore better forecasting complications, will led to a more efficient therapeutic plan in the concept of precision care.

## 5. Conclusions

The metabolic footprint revealed alterations in the metabolic milieu and mechanisms impairments independently associated with diabetes and its complications, detected throughout all classes of dysglycemia and in normoglycemia. This work is a proof of concept and future developments further integrating omics data will aid discovery of additional pathophysiological mechanisms in specific clusters. It is expected that deeper phenotyping and increased pathophysiology knowledge will allow to more precisely identify subjects affected in this multidimensional profile, predict their progression, and treat them accordingly towards precision medicine. We envisage that the metabolic footprint will assist a precision approach to type 2 diabetes

## Figures and Tables

**Figure 1 jcm-09-02588-f001:**
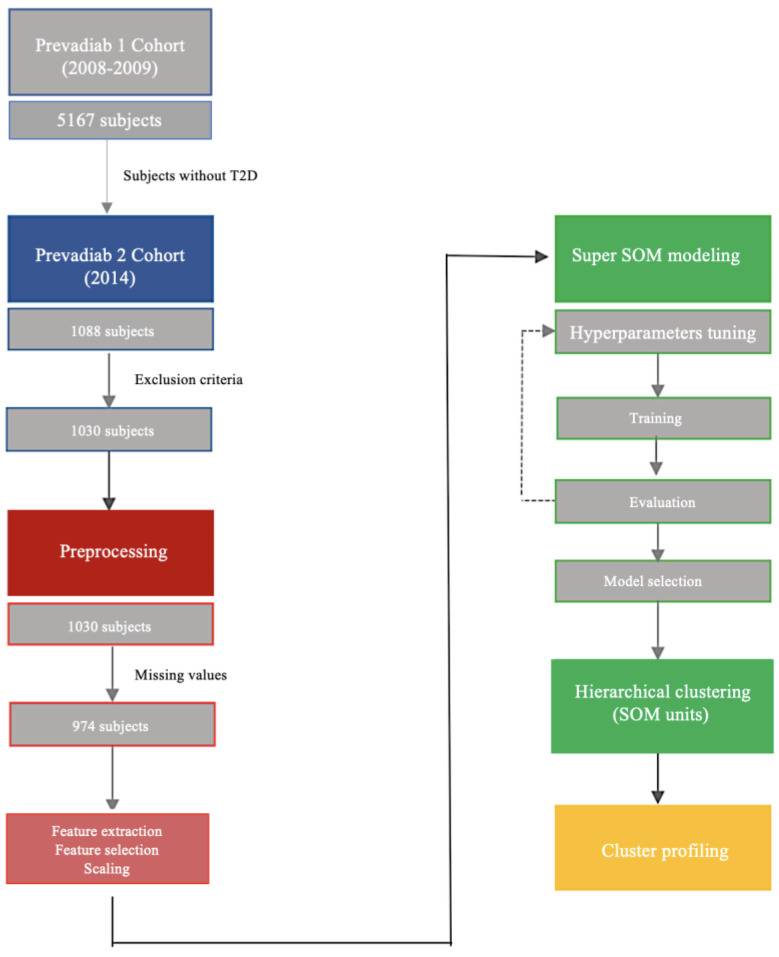
Methodology flowchart. In 2014, 1088 subjects from the PREVADIAB1 study without T2D in 2008–2009, were reevaluated 5 years later (PREVADIAB 2 cohort). After the application of exclusion criteria and preprocessing the data we performed a cluster analysis of 974 subjects. Finally, we profiled the clusters with several metabolic parameters. T2D-Type 2 Diabetes; SOM-Self-Organizing Map.

**Figure 2 jcm-09-02588-f002:**
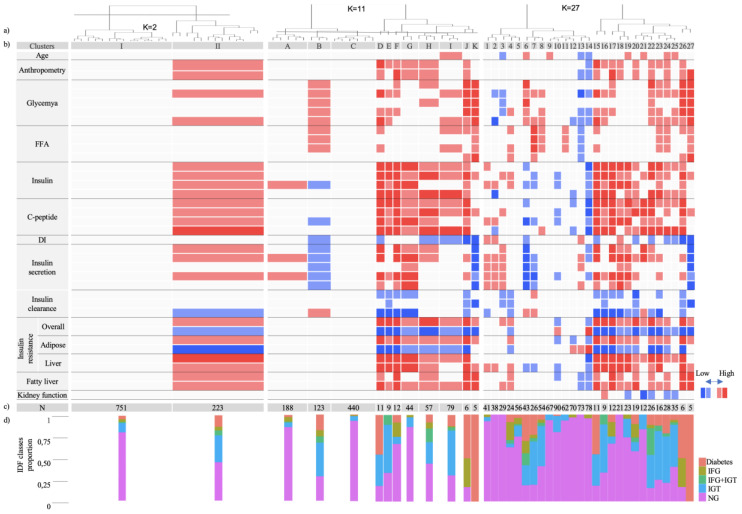
Clusters profiling. The dendogram of the hierarchical clustering of the 27 SOM units is shown for three cut levels of the dendogram (**a**): K = 2 (Cluster-I and -II); K = 11 (Clusters A–K); and K = 27 (Clusters 1–27). For each level, the clusters were profiled with several parameters, herein grouped by function, and the median of each cluster was compared with the distribution of the overall population (percentiles 10th, 25th, 75th, and 90th), and are displayed with a heatmap scale (**b**). Number of subjects in each cluster is shown in (**c**). Distribution of dysglycemia classes (IDF) in each cluster is shown in (**d**). DI-Disposition Index; FFA-Free fatty acids; IDF-International Diabetes Federation: IFG-Impaired fasting glycemia; IGT-Impaired glucose tolerance; NG-normoglycemia.

**Figure 3 jcm-09-02588-f003:**
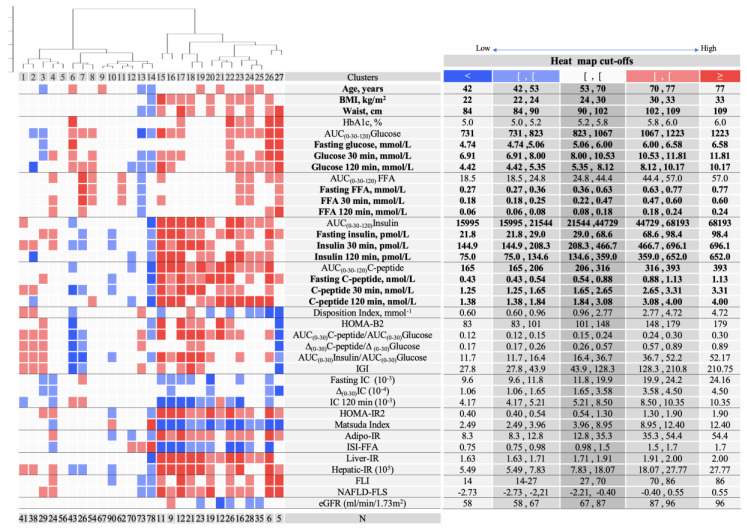
Clusters profiling for K = 27. Clusters were profiled with several parameters and the median of each cluster was compared with the distribution of the overall population (percentiles 10th, 25th, 75th, and 90th), and are displayed with a heatmap scale. The colors are defined by comparing the cluster median of the parameters with the distribution of the overall population (10th, 25th, 75th, and 90th percentiles). The table on the right shows the quantile limit values for each profiling parameter. Number of subjects in each cluster is displayed below the heatmap (N). AUC-Area under the curve; BMI-Body mass index; eGFR-CKD-estimated glomerular filtration rate using the Chronic Kidney Disease Epidemiology Collaboration (CKD-EPI) formula: FLI-Fatty liver Index; IC-Insulin clearance; IGI-Insulinogenic index; IR-Insulin resistance.

**Figure 4 jcm-09-02588-f004:**
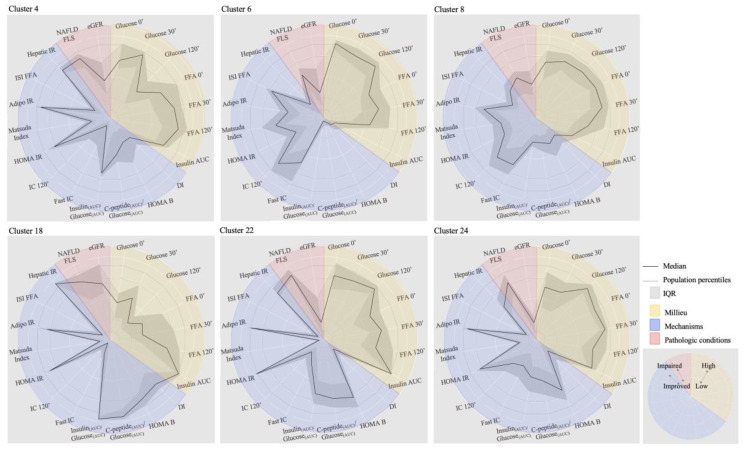
Metabolic footprint of Clusters 4, 6, 8, 18, 22, and 24 is shown in radar plots. The considered percentile ranges to profile the clusters (10th, 25th, 75th, and 90th) are represented in doted circumferences (light grey). Represented parameters are grouped in milieu (yellow), mechanisms (blue), and pathologic conditions (red). In the footprint higher deviations from the center in the milieu correspond to higher exposure, in mechanisms to more severe affection and to the presence and severity of pathological conditions. IQR-Interquartile range.

**Table 1 jcm-09-02588-t001:** Summary statistics of the population reported as median and (p25, p75) for continuous variables and as percentages for categorical variables. Parameters informing the cluster algorithm are in bold, the remaining parameters were used for cluster profiling. Comparison between genders was performed with Mann–Whitney test for continuous variables and with chi-square test for categorical variables.

Population	All		Male		Female		*p*-Value
**Total *n* (%)**	**974**		**387 (40)**		587 (60)		
Normoglycemia *n* (%) Prediabetes *n* (%) Diabetes (%)	707 (72) 211 (22) 56 (6)		279 (72) 85 (22) 23 (6)		428 (73) 126(21) 33 (6)		0.08
Parameters	Med	(p25, p75)	Med	(p25, p75)	Med	(p25, p75)	*p*-value
**Age, Years**	**62**	**(53,70)**	**62**	**(53, 70)**	**62**	**(53, 69)**	0.93
**BMI, kg/m^2^**	**27**	**(24, 30)**	**26**	**(25, 29)**	**27**	**(24, 31)**	0.020
**Waist, cm**	**96**	**(89, 102)**	**98**	**(92, 104)**	**94**	**(87, 101)**	<0.0001
HbA1c, %(mmol/mol)	5.5 (37)	(5.2, 5.8) (33, 40)	5.4 (36)	(5.2, 5.7) (33, 39)	5.5 (37)	(5.3, 5.8) (34, 40)	<0.0001
AUC_(0–30–120)_Glucose	928	(822, 1068)	927	(834, 1065)	929	(812, 1069)	0.78
**Fasting Glucose, mmol/L**	**5.46**	**(5.03, 5.96)**	**5.57**	**(5.14, 6.11)**	**5.40**	**(4.98, 5.89)**	<0.0001
**Glucose 30 min, mmol/L**	**9.16**	**(7.97, 10.54)**	**9.32**	**(8.11, 10.62)**	**9.05**	**(7.89, 10.44)**	0.023
**Glucose 120 min, mmol/L**	**6.58**	**(5.43, 8.26)**	**6.26**	**(5.09, 7.88)**	**6.74**	**(5.60, 8.37)**	0.0012
AUC_(0–30–120)_ FFA	33	(25, 44)	34	(26, 45)	35	(24, 44)	0.034
**Fasting FFA, mmol/L**	**0.48**	**(0.35, 0.63)**	**0.49**	**(0.36, 0.63)**	**0.47**	**(0.35, 0.63)**	0.34
**FFA 30 min, mmol/L**	**0.33**	**(0.24, 0.47)**	**0.33**	**(0.24, 0.47)**	**0.33**	**(0.25, 0.47)**	0.55
**FFA 120 min, mmol/L**	**0.12**	**(0.07, 0.17)**	**0.13**	**(0.09, 0.16)**	**0.11**	**(0.07, 0.16)**	<0.0001
AUC_(0–30–120)_Insulin	**31,225**	**(22, 136; 45, 501)**	**29,401**	**(19, 554; 42, 928)**	**32,019**	**(4653, 23,534)**	0.002
**Fasting insulin, pmol/L**	**43.1**	**(29.1, 68.8)**	**42.4**	**(28.3, 68.4)**	**43.8**	**(29.8, 68.8)**	0.35
**Insulin 30 min, pmol/L**	**318**	**(210, 468)**	**300**	**(194, 464)**	**331**	**(223, 469)**	0.029
**Insulin 120 min, pmol/L**	**238**	**(138, 379)**	**210**	**(112, 317)**	**252**	**(158, 401)**	<0.0001
AUC_(0–30–120)_C-peptide	256	(208, 318)	251	(202, 311)	261	(202, 322)	0.043
**Fasting C-peptide, nmol/L**	**0.68**	**(0.54, 0.88)**	**0.68**	**(0.54, 0.89)**	**0.67**	**(0.54, 0.86)**	0.57
**C-peptide 30 min, nmol/L**	**2.09**	**(1.66, 2.65)**	**2.09**	**(1.64, 2.63)**	**2.11**	**(1.66, 2.66)**	0.57
**C-peptide 120 min, nmol/L**	**2.59**	**(1.91, 3.47)**	**2.48**	**(1.77, 3.38)**	**2.70**	**(2.03, 3.53)**	0.003
Disposition Index, mmol^−1^	1.67	(0.98, 2.83)	1.61	(0.90, 2.67)	1.72	(1.01, 2.88)	0.14
HOMA-B2	123	(102, 148)	118	(46, 98)	118	(105, 150)	0.003
AUC_(0–30)_ C-peptide/ AUC_(0–30)_ Glucose	0.19	(0.15, 0.24)	0.19	(0.14, 0.24)	0.20	(0.15, 0.25)	0.07
∆_(0–30)_C-peptide/ ∆_(0–30)_Glucose	0.38	(0.26, 0.57)	0.37	(0.25, 0.55)	0.38	(0.26, 0.58)	0.29
AUC_(0–30)_ Insulin/AUC_(0–30)_ Glucose	25.6	(16.7, 37.1)	23.5	(14.5, 35.7)	27.1	(18.3, 37.8)	0.004
IGI	77.0	(45.9, 129.4)	73.3	(40.3, 121.1)	79.9	(47.6, 135.5)	0.018
Fasting IC (10^−3^)	15.2	(11.8, 19.7)	15.8	(12.1, 20.4)	14.9	(11.6, 19.3)	0.0199
∆_(0–30)_IC (10^−4^)	2.43	(1.65, 3.56)	2.54	(1.69, 3.65)	2.39	(1.62, 3.51)	0.32
IC 120 min (10^−3^)	6.69	(5.20, 8, 35)	7.05	(5.25, 8.89)	6.54	(5.16, 8.12)	0.005
HOMA-IR2	0.81	(0.55, 1.30)	0.80	(0.53, 1.30)	0.82	(0.56, 1.28)	0.48
Matsuda Index	6.14	(3.85, 8.77)	6.27	(4.06, 9.14)	6.11	(3.78, 8.62)	0.22
Adipo-IR	20.9	(12.9, 35.1)	20.5	(12.6, 35.6)	21.3	(13.0, 35.0)	0.76
ISI-FFA	1.14	(0.90, 1.36)	1.15	(0.88, 1.37)	1.12	(0.90, 1.35)	0.45
Liver-IR	1.81	(1.73, 1.91)	1.79	(1.69,1.90)	1.83	(1.74, 1.93)	<0.0001
Hepatic-IR (10^5^)	11.9	(7.8, 18.0)	11.6	(7.5, 18.3)	12.1	(8.1, 17.9)	0.35
FLI	47	(25, 69)	55	(33, 75)	40	(20, 64)	<0.0001
NAFLD-FLS	–1.30	(−2.22, −0.42)	−1.29	(−2.18, −0.27)	−1.32	(−2.24, −0.52)	0.38
eGFR-CKD (mL/min/1.73 m^2^)	77	(66, 87)	80	(69, 89)	75	(65, 85)	<0.0001

AUC-Area under the curve; BMI-Body Mass Index; eGFR-CKD-estimated glomerular filtration rate using the Chronic Kidney Disease Epidemiology Collaboration (CKD-EPI) formula; FFA-Free fatty acids; IC-Insulin clearance; IGI-Insulinogenic index; IR-Insulin resistance; HbA1c-Hemoglobin A1c.

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
