# Peer review of "Metabolic Footprint, towards Understanding Type 2 Diabetes beyond Glycemia"

_jcm, 2020, doi:10.3390/jcm9082588_

Round 1
Reviewer 1 Report
This is a valuable work as it presents an original perspective on diabetes pathogenesis and diabetes complications. Changes in metabolic milieu other than glycemic imbalance constitute new and promising direction in research on type 2 diabetes. High number of parameters taken into consideration in the study makes the Results section a bit unclear, but, as Authors admit, it is difficult to describe the complexity of this problem in an understandable manner without simplifying it. However, in my opinion the Authors should attempt to rewrite the Results section in a less complicated way. What is more, sentence "This work reveals that T2D heterogeneity can be resolved by a thorough metabolic milieu, beyond glycemia, and mechanisms profiling – metabolic footprint" is not correct.
Author Response
Dear Reviewer,
We thank the Reviewer for his/her valuable feedback and comments. In what follows, we address in detail all of them.
This is a valuable work as it presents an original perspective on diabetes pathogenesis and diabetes complications. Changes in metabolic milieu other than glycemic imbalance constitute new and promising direction in research on type 2 diabetes. High number of parameters taken into consideration in the study makes the Results section a bit unclear, but, as Authors admit, it is difficult to describe the complexity of this problem in an understandable manner without simplifying it. However, in my opinion the Authors should attempt to rewrite the Results section in a less complicated way.
We thank the Reviewer for recognizing the value and novelty of the work, which we believe is paramount to the understanding of type 2 diabetes.
After addressing the Result section in several different ways, we agreed that we have reached what we thought was a simplified presentation of the results. We would really appreciate if the Reviewer could suggest to us how we could improve this section.
What is more, sentence "This work reveals that T2D heterogeneity can be resolved by a thorough metabolic milieu, beyond glycemia, and mechanisms profiling – metabolic footprint" is not correct.
We agree with the Reviewer and have changed the sentence to “This work reveals that T2D heterogeneity can be captured by a thorough metabolic milieu and mechanisms profiling – metabolic footprint” (line 29-31).
Sincerely,
M Paula Macedo
Reviewer 2 Report
Authors investigated the stratifying a population regarding the metabolic milieu, including 974 subjects with normoglycemia, prediabetes or non-treated diabetes using a cluster analysis to clarify chief pathophysiological mechanisms involved in dysmetabolism. In this study, two clusters (C-I and C-II) were identified. C-I mainly included representing normoglycemia. On the other hand, C-II showed higher BMI, waist circumference, glycemia, insulinemia and C-peptide levels, and therefore is associated with insulin resistance and fatty liver. However, both clusters contained a significant number of subjects with dysglycemia. This study indicated that the standard clinical evaluation of fasting and post-prandial glycemia does not reflect the overall exposure to glucose or express the metabolic milieu heterogeneity. These results are of significance to clinical practice. There are, however, some points which should be addressed.
Specific comments
Criticism 1: (Supplementary Table1) “AUC(0-30)C-peptide/AUC(0-30)Glucose and AUC(0-30)Insulin/AUC(0-30)Glucose” Please show the references.
Criticism 2: (Methods) “In 2014, we randomly selected 84 within this cohort 1,088 subjects that did not have diabetes in PREVADIAB1”. (Results) “We analyzed 974 subjects of the PREVADIAB2”: Authors should show the flowchart of these subjects.
Criticism 3: (Methods) Line 119: “Glucose0min+Glucose 0min” Is it true?
Author Response
Dear Reviewer,
We thank the Reviewer for his/her valuable feedback and comments. In what follows, we address in detail all of them.
Authors investigated the stratifying a population regarding the metabolic milieu, including 974 subjects with normoglycemia, prediabetes or non-treated diabetes using a cluster analysis to clarify chief pathophysiological mechanisms involved in dysmetabolism. In this study, two clusters (C-I and C-II) were identified. C-I mainly included representing normoglycemia. On the other hand, C-II showed higher BMI, waist circumference, glycemia, insulinemia and C-peptide levels, and therefore is associated with insulin resistance and fatty liver. However, both clusters contained a significant number of subjects with dysglycemia. This study indicated that the standard clinical evaluation of fasting and post-prandial glycemia does not reflect the overall exposure to glucose or express the metabolic milieu heterogeneity. These results are of significance to clinical practice. There are, however, some points which should be addressed.
We thank the Reviewer for recognizing the significance of this work for the future management of subjects with Type 2 Diabetes.
Specific comments
Criticism 1: (Supplementary Table1) “AUC(0-30)C-peptide/AUC(0-30)Glucose and AUC(0-30)Insulin/AUC(0-30)Glucose” Please show the references.
We added now the references for the above-mentioned formulas and also a Reference section in Supplementary material file.
Criticism 2: (Methods) “In 2014, we randomly selected 84 within this cohort 1,088 subjects that did not have diabetes in PREVADIAB1”. (Results) “We analyzed 974 subjects of the PREVADIAB2”: Authors should show the flowchart of these subjects.
We agree with the Reviewer and therefore included Figure1 – Methodology Flowchart.
Criticism 3: (Methods) Line 119: “Glucose0min+Glucose 0min” Is it true?
We thank the Reviewer for detecting the typing error. It was not correct, and it has been changed to “Glucose0min+Glucose30min” (line125).
Sincerely,
M Paula Macedo